# Topological-Aware Regularization for Semi-Supervised Intracranial Aneurysm Vessel Segmentation

**Feiyang Xiao**[1,2,*]     FYXIAO24@M.FUDAN.EDU.CN

**Yichi Zhang**[1,2,*]     ZHANGYICHI23@M.FUDAN.EDU.CN

**Xigui Li**[1,2]     LIXIGUI@FUDAN.EDU.CN

**Yuanye Zhou**[2,3]     ZHYY2009@163.COM

**Chen Jiang**[1,2]     JIANGCHEN@SAIS.COM.CN

**Xin Guo**[1,2]     GUOXIN@SAIS.COM.CN

**Limei Han**[1,2]     HANLIMEI@FUDAN.EDU.CN

**Yuxin Li**[4,†]     LIYUXIN@FUDAN.EDU.CN

**Fengping Zhu**[4,†]     ZHUFENGPING@FUDAN.EDU.CN

**Yuan Cheng**[1,2,†]     CHENG_YUAN@FUDAN.EDU.CN

[1] *Artificial Intelligence Innovation and Incubation Institute, Fudan University, Shanghai, China*

[2] *Shanghai Academy of Artificial Intelligence for Science, Shanghai, China*

[3] *Hong Kong Polytechnic University, Hong Kong*

[4] *Huashan Hospital, Fudan University, Shanghai, China*

**Editors:** Accepted for publication at MIDL 2026

## Abstract

Accurate segmentation of intracranial aneurysm and their parent vessels (IA-Vessel) from magnetic resonance angiography is a critical prerequisite for computational fluid dynamics-based rupture risk assessment. While deep learning methods can automate this laborious task, they are hindered by the high cost and scarcity of expert annotations. Most existing semi-supervised methods focus on enforcing regional constraints while largely ignoring topological constraints, which is insensitive to subtle but critical errors like vessel adhesion or surface irregularities, which are often unsuitable for downstream applications. To address this gap, we introduce topological-aware regularization (TAR) by incorporating the learning of local vascular topology to ensure the precise and geometrically correct segmentation of the IA-Vessel complex using only a small amount of labeled data. Experimental results on a multi-center MRA dataset show that our framework efficiently utilizes unlabeled data and outperforms state-of-the-art semi-supervised segmentation methods. Instead of being restricted to a fixed framework, TAR is a plug-and-play strategy that can be seamlessly integrated into various semi-supervised frameworks to further boost their performance. Code is available at https://github.com/AbsoluteResonance/TAR.

**Keywords:** Intracranial Aneurysm Segmentation, Semi-Supervised Learning, Topological-Aware Regularization

---

[*] Contributed equally

[†] Corresponding authors.

## 1. Introduction

Intracranial aneurysm (IA) is a pathological dilation of blood vessels, primarily occurring at arterial bifurcations (Schievink, 1997). Although often initially asymptomatic, IAs can enlarge and rupture, leading to subarachnoid hemorrhage, which is associated with severe morbidity and mortality (Cebral et al., 2005). Consequently, the accurate assessment of rupture risk is essential for guiding clinical intervention (Etminan and Rinkel, 2016). Computational Fluid Dynamics (CFD) has emerged as a vital tool in this domain, offering biomechanical insights by quantifying hemodynamic parameters such as wall shear stress and oscillatory shear index (Li et al., 2025; Morris et al., 2016; Wang et al., 2025).

Magnetic resonance angiography (MRA) serves as a high-resolution, non-invasive imaging modality for visualizing the detailed anatomical features of aneurysms (Pierot et al., 2013). To perform CFD analysis, an accurate segmentation of the Intracranial Aneurysm and its Parent Vessels (IA-Vessel) from these images is a critical prerequisite (Patel et al., 2023). Traditionally, this requires manual segmentation by radiologists, which is a labor-intensive and time-consuming procedure subject to inter-observer variability. While Deep Learning has emerged as a state-of-the-art approach for automating this task (Antonelli et al., 2022; Ma et al., 2022), its performance relies heavily on the availability of large-scale, pixel-level annotated datasets.

However, acquiring extensively annotated medical datasets is particularly challenging for cerebrovascular structures. The annotation process requires high-level domain expertise to distinguish complex vascular connectivity from background noise, making it expensive and resource-intensive (Tajbakhsh et al., 2020; Shi et al., 2024). Given the abundance of unlabeled clinical data compared to scarce labeled examples, Semi-Supervised Learning (SSL) presents an attractive solution (Jiao et al., 2023). By leveraging a limited set of labeled data alongside a large volume of unlabeled data, SSL aims to reduce the annotation burden while maintaining high performance.

Despite the promise of SSL, applying standard semi-supervised methods to IA-Vessel segmentation presents unique challenges. Most existing SSL frameworks rely on region overlap-based consistency constraints (e.g., Dice or MSE loss) to propagate information from labeled to unlabeled data. These metrics focus on volumetric accuracy but are insensitive to the topological integrity of vascular networks. In the context of CFD, even subtle topological errors such as vessel adhesion (fusion of adjacent arteries) or breaks in connectivity can render the segmentation useless, leading to mesh generation failures or severe flow field distortions (Xiao et al., 2025).

To address this gap, we introduce Topological-Aware Regularization (TAR) for semi-supervised intracranial aneurysm vessel segmentation. Unlike previous approaches that focus solely on regional consistency, our method explicitly incorporates local vascular topology constraints into the semi-supervised learning process. By enforcing the structural integrity of the vessel skeleton and centerline, TAR ensures that the model learns geometrically correct segmentations even with limited supervision.

## 2. Related Work

### 2.1. Intracranial Aneurysm and Vessel Segmentation

The segmentation of intracranial aneurysms and vessels has evolved from traditional image processing techniques to advanced deep learning methodologies. Early approaches relied heavily on vesselness-based filters, most notably the multiscale Hessian filter (Frangi et al., 1998). While effective for clear tubular structures, these methods often struggle with the complex, irregular geometries of aneurysm sacs and varying vessel diameters, lacking the geometric fidelity required for downstream hemodynamic analysis (Lamy et al., 2022).

With the advent of deep learning, Convolutional Neural Networks (CNNs) have become the standard. General-purpose medical segmentation frameworks, such as 3D U-Net (Çiçek et al., 2016) and the self-configuring nnU-Net (Isensee et al., 2021), prioritize global voxel-wise accuracy. While they achieve high Dice scores, they often lack specific mechanisms for preserving the connectivity of fine vessel branches or accurately delineating the aneurysm neck. To address these limitations, task-specific architectures have been proposed. Glia-Net (Bo et al., 2021) utilizes global context fusion to enhance aneurysm delineation, while detection-based frameworks like nnDetection (Baumgartner et al., 2021) and sphere-based detectors like CPM-Net (Song et al., 2020) focus on robust localization. More recently, AA-Seg (Yao et al., 2024) pioneered joint aneurysm-vessel segmentation.

However, a persistent limitation across these supervised methods is their reliance on region-based loss functions (e.g., Dice, Cross-Entropy). These metrics are insensitive to topological abnormalities. As noted in recent benchmarks (Xiao et al., 2025), high volumetric overlap does not guarantee topological correctness; models frequently produce vessel adhesions or disconnects that invalidate CFD simulations. While recent works have explored topological loss functions like clDice (Shit et al., 2021) in fully supervised settings, integrating such geometric constraints into label-scarce, semi-supervised regimes remains an unexplored frontier.

### 2.2. Semi-Supervised Learning in Medical Imaging

Semi-supervised learning has garnered significant attention in medical imaging as a solution to the scarcity of pixel-level annotations. The evolution of SSL methodologies can be broadly categorized into adversarial learning, consistency regularization, and pseudo-labeling frameworks.

Early approaches adapted generative adversarial networks to align the data distributions of labeled and unlabeled sets. A representative method is ADV (Hung et al., 2018), which employs a discriminator network to encourage the segmentation model to produce predictions on unlabeled data that are indistinguishable from ground-truth labels. Similarly, Entropy Minimization methods (Vu et al., 2019), drive the network to produce high-confidence predictions by minimizing the entropy of the output probability maps, thereby pushing decision boundaries away from high-density regions.

Consistency regularization has subsequently emerged as the dominant paradigm, positing that a model's predictions should remain invariant to perturbations. The Mean Teacher framework (Tarvainen and Valpola, 2017) established a robust baseline by enforcing consistency between a student model and a temporally averaged teacher model. Building on this, UAMT (Yu et al., 2019) integrates uncertainty estimation, filtering out unreliable predictions from the consistency loss to improve training stability. To further explore the data manifold, Interpolation Consistency Training (Verma et al., 2022) enforces consistency at interpolated points between unlabeled samples, encouraging a smoother decision boundary.

To mitigate the confirmation bias inherent in single-model approaches, dual-network architectures with cross-supervision have been developed. Cross Pseudo Supervision (Chen et al., 2021) trains two networks with different initializations and uses the one-hot pseudo-labels from one network to supervise the other. Advanced variants have since focused on refining the quality of these supervisory signals. For instance, UGMCL (Zhang et al., 2023) introduces uncertainty-guided mutual consistency to weight the loss based on prediction reliability, while ACMT (Xu et al., 2023) specifically targets ambiguous regions in the teacher's predictions. More recent works like CML (Wu et al., 2024) and RD (Wu et al., 2021) further enhance robustness against noise in pseudo-labels through cross-model learning and regularized dropout strategies, respectively. The state-of-the-art framework, DyCON (Assefa et al., 2025), pushes this direction further by combining uncertainty-aware consistency with contrastive learning to learn more discriminative feature representations.

Despite these rapid advancements, a fundamental limitation persists: virtually all aforementioned methods optimize region-based objectives (e.g., Dice, Cross-Entropy, or MSE). These metrics prioritize volumetric overlap but are mathematically insensitive to topological properties. In the context of IA-Vessel segmentation, this allows models to achieve competitive Dice scores while failing to preserve critical connectivity, resulting in broken vessel skeletons or fused arteries that invalidate downstream CFD analysis. Our work addresses this specific gap by introducing explicit topological constraints into the semi-supervised learning process.

## 3. Method

### 3.1. Task Definition

In clinical workflows, analyzing high-resolution MRA volumes typically follows a two-stage pipeline: global detection of lesions followed by localized fine-grained segmentation. This study specifically targets patch-based segmentation, as pixel-level annotation represents the primary bottleneck in model development compared to the less intensive bounding-box annotations required for detection. Therefore, our annotation-efficient patch-based segmentation model is designed to be integrated with detection pre-processors to achieve a practical, end-to-end solution for full MRA volumes.

Furthermore, the segmentation target is strictly defined by the requirements of downstream Computational Fluid Dynamics analysis. For accurate rupture risk assessment, segmenting the aneurysm sac in isolation is insufficient. To construct accurate hemodynamic boundary conditions, it is essential to fully segment the Intracranial Aneurysm-Vessel complex, including the connected parent arteries.

With this clinical context established, we formally define the semi-supervised segmentation task. Given a training dataset $D$, it is split into a labeled set with $M$ cases, denoted as $D_L = \{x_i^l, y_i\}_{i=1}^M$, and an unlabeled set with $N$ cases, denoted as $D_U = \{x_i^u\}_{i=1}^N$. Here, $x^l$ and $x^u$ represent the input patch images, and $y_i$ is the corresponding ground-truth segmentation for the labeled data. The model is required to utilize both $D_L$ and $D_U$ during the training phase, enabling the network to produce segmentation results for new images during inference that are comparable to those of an optimal model trained on a fully labeled dataset.

To accomplish this, semi-supervised learning is typically designed as a two-fold task. First, a supervised loss is applied to the labeled set $D_L$, similar to fully-supervised methods, to ensure the network effectively learns features from the available labels. Second, an unsupervised regularization term is introduced for the unlabeled set $D_U$. For example, consistency regularization aims to penalize differences in predictions for the same input under various perturbations. By doing so, it forces the network to maintain stable predictions against disturbances in the input space, which, in turn, smoothly propagates label information from labeled to unlabeled regions

### 3.2. Semi-Supervised Backbone

The Mean Teacher framework (Tarvainen and Valpola, 2017) is widely used in semi-supervised image segmentation. It consists of a student model and a teacher model, which share an identical structure but employ different parameter update strategies. During training, labeled data is fed into the student model, and a supervised loss is calculated between its output and the ground-truth labels. In contrast, the teacher model is updated by taking an Exponential Moving Average (EMA) of the student model's weights during the training stage as follows.

$$\theta_t = \mu\theta_t + (1 - \mu)\theta_s \tag{1}$$

where $\theta_t$ and $\theta_s$ are the parameters of the teacher model and the student model, and $\mu$ is a momentum coefficient. This process makes the teacher model a more robust and reliable source of pseudo-labels, as it averages out the rapid fluctuations of the student's training process. This approach allows the teacher model to provide a more stable and progressively refined target distribution throughout the training process. For unlabeled data, it is passed through different augmentations or perturbations and then fed into both the Teacher and student models separately. This process yields two sets of prediction probability maps, and a consistency loss is calculated between them to facilitate learning from the unlabeled data.

Building upon the teacher-student architecture, more recent semi-supervised approaches like DyCON (Assefa et al., 2025) introduces additional Uncertainty-aware Consistency Loss (UnCL) and the Focal Entropy-aware Contrastive Loss (FeCL). At a global scale, UnCL integrates voxel-wise uncertainty directly into the consistency loss via an entropy-driven dynamic weighting mechanism. While these general-propose semi-supervised methods have demonstrated further advancements on many segmentation benchmarks, their success is often measured on the segmentation of well-defined organs, causing them to overlook the critical topological structures required for more complex tasks like IA-Vessel segmentation. This issue is compounded because most existing models are evaluated on region overlap-based regularization. This metric is insensitive to geometric and topological abnormalities such as vessel adhesion and surface irregularities. Consequently, this often results in segmentation outcomes that are unsuitable for downstream applications due to subsequent mesh generation failures or flow field distortions.

### 3.3. Topological-Aware Regularization

Vascular networks are fundamentally tubular structures with complex topological properties, where connectivity and branching patterns represent their core anatomical features. However, existing semi-supervised segmentation methods primarily leverage unlabeled data through pixel-level or feature-level consistency, often neglecting this complex structural information. To address this challenge, we propose a plug-and-play topology-aware regularization loss, denoted as $L_{Topo}$, to enhance the model's awareness of structural integrity. The $L_{Topo}$ is composed of a weighted sum of two complementary loss functions to optimize the vessel's topological structure from the perspectives of centerline matching with $\mathcal{L}_{clDice}$ and skeleton integrity $\mathcal{L}_{Skel}$ as follows:

$$\mathcal{L}_{\text{Topo}} = \lambda_1 \underbrace{\left( -\frac{2\sum_i \mathcal{T}(p_i)\mathcal{T}(g_i)}{\sum_i \mathcal{T}(p_i) + \sum_i \mathcal{T}(g_i)} \right)}_{\mathcal{L}_{\text{clDice}}} + \lambda_2 \underbrace{\left( -\frac{\sum_i p_i \mathcal{S}(g_i)}{\sum_i g_i} \right)}_{\mathcal{L}_{\text{Skel}}} \tag{2}$$

where $p_i$ is the predicted probability at voxel $i$ from the student model and $g_i$ is the pseudo-label at voxel $i$ from the teacher model. $\lambda_1, \lambda_2$ are weighting coefficients that balance the relative importance of the two component losses. In our following experiment, we set both $\lambda_1$ and $\lambda_2$ to 1. $\mathcal{S}(\cdot)$ represents the soft-skeletonization function, which takes a probability map and outputs a map highlighting the central skeleton of the structure. $\mathcal{T}(\cdot)$ calculates the centerline probability map as follows.

$$\mathcal{T}(p_i) = \frac{\sum_i g_i \mathcal{S}(p_i)}{\sum_i \mathcal{S}(p_i)}, \quad \mathcal{T}(g_i) = \frac{\sum_i p_i \mathcal{S}(g_i)}{\sum_i \mathcal{S}(g_i)} \tag{3}$$

In semi-supervised framework, the binarized pseudo-labels $g$ generated by the teacher model are treated as masks to supervise the student model's predictions $p$.

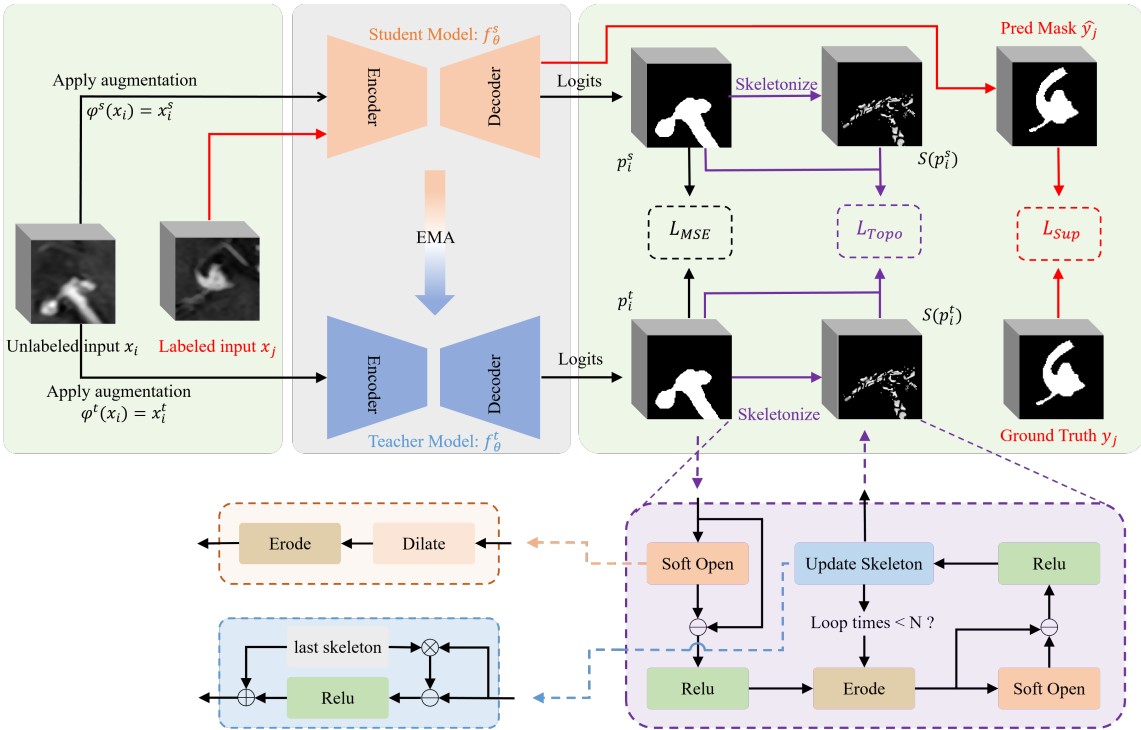

Figure 1: Overview of our proposed Topological-Aware Regularization framework for semi-supervised intracranial aneurysm vessel segmentation framework, the purple $\mathcal{L}_{\text{Topo}}$ and the purple Skeletonize in the figure are our core modules. For labeled data, a standard supervised segmentation loss ($\mathcal{L}_{\text{sup}}$) is applied to the student model's predictions. For unlabeled data, the teacher model's outputs are used as pseudo-labels to compute both the standard consistency loss ($\mathcal{L}_{\text{MSE}}$) and the topology-aware regularization loss ($\mathcal{L}_{\text{Topo}}$) that we propose. The figure also details our differentiable soft skeletonization algorithm, which approximates morphological erosion and dilation operations through a series of min-pooling and max-pooling operations. The $\mathcal{L}_{\text{Topo}}$ loss incorporates $\mathcal{L}_{\text{clDice}}$ and $\mathcal{L}_{\text{Skel}}$, which jointly penalize topological discrepancies between the student model's predictions and the teacher-generated pseudo-labels.

The original skeleton extraction algorithm is CPU-based (Kirchhoff et al., 2024). While its performance is sufficient in fully-supervised settings, this approach becomes computationally prohibitive for semi-supervised tasks where the teacher's pseudo-labels must be generated in real-time. To overcome this limitation, we adopted the differentiable soft-skeletonization as proposed in the clDice (Shit et al., 2021), to create a unified generation

process for the skeletons required by both regularization losses. This algorithm efficiently approximates the vessel centerline skeleton on the GPU via a series of max-pooling and min-pooling operations, while maintaining full differentiability. Consequently, this unified design not only mitigates computational overhead but also ensures that both topological constraints operate on a consistent structural representation, thereby enhancing the model's capacity for learning both vessel connectivity and structural integrity.

### 3.4. Overall Training Procedure

The overall training objective of our proposed framework is to minimize the weighted sum of supervised segmentation loss $\mathcal{L}_{\mathrm{sup}}$, unsupervised regularization loss of semi-supervised backbone $\mathcal{L}_{\mathrm{unsup}}$ and our proposed topological-aware regularization $\mathcal{L}_{\mathrm{Topo}}$ as follows.

$$\mathcal{L} = \mathcal{L}_{\mathrm{sup}} + \mathcal{L}_{\mathrm{unsup}} + \mathcal{L}_{\mathrm{Topo}} \tag{4}$$

## 4. Experiments

### 4.1. Datasets

We conduct extensive experiments on the Intracranial Aneurysm Vessel Segmentation (IAVS) dataset (Xiao et al., 2025), which contains multi-center collection of 641 high-resolution 3D MRA images. In total, 587 IAs and their corresponding parent vessels were annotated and selected out to form patch volumes for training and validation of the segmentation framework. The topological integrity of every vessel is guaranteed during the annotation procedure, providing a gold standard for validating the topology-preserving capabilities of segmentation models. We randomly partitioned the dataset into 357 cases for training, 99 cases for validation, and 66 cases for final testing evaluation. In our experiments, we use 5%, 10% and 20% of the training set (17, 35, and 71 cases) as labeled data, while the remaining cases served as unlabeled data, for which only the images were used during training. We compared our method against a series state-of-the-art semi-supervised segmentation methods. All methods were trained and evaluated under the identical labeled data configuration to ensure a fair comparison.

### 4.2. Implementation Details

All of our experiments are implemented in Python with PyTorch, using an NVIDIA A100 GPU. The backbone segmentation network for the specialist model is 3D U-NET (Çiçek et al., 2016). We use the SGD optimizer with an initial learning rate of 0.01, a weight decay of 1e-4 and a momentum of 0.9 to update the network parameters with the maximum iteration number set to 10000. This network employs a standard 3D U-Net architecture, featuring four downsampling and four upsampling operations, with encoder channel counts of 64, 128, 256, 512, and 1024, respectively. During the decoding phase, skip connections

are used for upsampling, and Dropout layers with a dropout rate of 0.3 are added before the bottleneck layer and the final output to prevent overfitting.

To quantitatively evaluate the performance of all methods, we employed standard metrics to assess segmentation accuracy from different perspectives. Dice Similarity Coefficient (Dice) is used to measure the volumetric overlap between the predicted segmentation and the ground truth. To quantify the discrepancy in volume, we use the Relative Absolute Volume Difference (RAVD). Furthermore, we evaluate the surface-to-surface accuracy using the Average Surface Distance (ASD) and the 95th percentile of the Hausdorff Distance (95HD). The ASD measures the average distance between the boundaries of the predicted and ground truth objects, while the 95HD provides a more robust measure of the maximum surface distance by excluding the top 5% of outlier distances. Finally, to evaluate the topological preservation of the vascular networks, we employ the clDice (Shit et al., 2021), which measures the overlap between the extracted skeletons of the prediction and the ground truth.

### 4.3. Comparison Experiments

Table. 1 presents the performance of our method with comparison to other representative semi-supervised frameworks (Hung et al., 2018; Chen et al., 2021; Wu et al., 2021; Xu et al., 2023; Verma et al., 2022; Yu et al., 2019; Vu et al., 2019; Zhang et al., 2023; Wu et al., 2024) using different numbers of labeled images. An important observation from the results is the limited efficacy of several classic semi-supervised methods when applied to vessel segmentation. Notably, several methods such as ADV and CPS perform even worse than the Supervised Baseline trained with only the labeled data. This phenomenon highlights a critical challenge that methods that rely purely on region-based consistency are ill-suited for tasks where topological integrity is paramount. These conventional approaches enforce consistency at the pixel or patch level, which can be counterproductive for vessel networks and may penalize small valid gaps between different vessel segments. As a result, they may introduce misleading supervisory signals that corrupt the learning process.

In contrast, building upon the vanilla mean teacher framework (Tarvainen and Valpola, 2017), utilizing TAR consistently enhances the performance in all annotation scenarios. To further validate its impact on the leading edge of current research, we conduct experiments on DyCON (Assefa et al., 2025), a recently proposed state-of-the-art semi-supervised framework. As demonstrated in the table, the addition of TAR provides further performance increase and establishes a new state-of-the-art, showcasing its power as a plug-and-play module. This initial finding confirms that our topology-aware module provides a substantial and meaningful improvement to established semi-supervised methods. From the visualization of segmentation results in Figure. 2 We can observe that our proposed method generates more accurate predictions compared with other methods, which further demonstrates the effectiveness of our proposed method.

Table 1: Comparative experimental results between our proposed method and other semi-supervised segmentation methods on IAVS dataset with 5%, 10% and 20% annotation settings.

| Method | Annotation | Dice [%] | clDice [%] | RAVD [%] | ASD[voxel] | 95HD[voxel] |
|---|---|---|---|---|---|---|
| Supervised Baseline | 5% | 66.03 ± 9.78 | 67.28 ± 14.68 | 39.83 ± 21.60 | 1.79 ± 0.76 | 37.70 ± 27.30 |
| ADV (BMVC'18) | 5% | 64.10 ± 12.22 | 63.36 ± 16.04 | 39.10 ± 20.95 | 1.72 ± 0.78 | 36.13 ± 28.18 |
| CPS (CVPR'21) | 5% | 64.37 ± 9.14 | 69.81 ± 13.53 | 45.97 ± 22.88 | 1.63 ± 0.75 | 28.70 ± 17.92 |
| RD (NeuriPS'21) | 5% | 65.70 ± 9.91 | 66.02 ± 14.33 | 38.48 ± 21.67 | 1.69 ± 0.74 | 32.82 ± 24.34 |
| ACMT (MedIA'23) | 5% | 64.72 ± 11.13 | 65.39 ± 14.54 | 47.31 ± 32.23 | 1.74 ± 0.74 | 36.83 ± 27.74 |
| ICT (NN'22) | 5% | 65.51 ± 11.86 | 65.23 ± 15.22 | 40.60 ± 20.47 | 1.72 ± 0.80 | 33.88 ± 26.00 |
| UAMT (MICCAI'19) | 5% | 66.15 ± 10.73 | 69.22 ± 15.52 | 39.91 ± 21.93 | 1.67 ± 0.78 | 35.35 ± 30.43 |
| EM (CVPR'19) | 5% | 65.02 ± 11.67 | 66.63 ± 14.81 | 39.92 ± 20.99 | 1.71 ± 0.76 | 33.96 ± 24.72 |
| UGMCL (AIIM'23) | 5% | 65.54 ± 13.47 | 66.14 ± 16.26 | 39.23 ± 27.74 | 1.74 ± 0.76 | 37.48 ± 32.69 |
| CML (ACMMM'24) | 5% | 63.85 ± 14.90 | 68.29 ± 16.61 | 39.96 ± 21.39 | 1.87 ± 0.87 | **15.77** ± 16.80 |
| MT (NeuriPS'17) | 5% | 66.14 ± 10.82 | 67.81 ± 15.14 | 39.32 ± 20.26 | 1.76 ± 0.77 | 34.80 ± 25.30 |
| **MT + TAR** | 5% | 67.99 ± 9.75 | 68.02 ± 14.60 | 42.05 ± 26.60 | 1.69 ± 0.75 | 38.92 ± 29.07 |
| DyCON (CVPR'25) | 5% | 67.40 ± 11.40 | 67.86 ± 15.31 | 40.26 ± 25.14 | 1.66 ± 0.84 | 20.85 ± 17.61 |
| **DyCON + TAR** | 5% | **70.77** ± 13.59 | **74.04** ± 13.12 | **33.35** ± 23.01 | **1.48** ± 0.89 | 23.55 ± 25.42 |
| Supervised Baseline | 10% | 66.64 ± 10.86 | 67.70 ± 13.97 | 41.02 ± 24.16 | 1.67 ± 0.76 | 30.95 ± 25.06 |
| ADV (BMVC'18) | 10% | 64.72 ± 12.08 | 65.74 ± 14.68 | 42.27 ± 24.86 | 1.66 ± 0.74 | 28.16 ± 23.47 |
| CPS (CVPR'21) | 10% | 67.38 ± 12.63 | 68.50 ± 16.72 | 34.94 ± 20.07 | 1.70 ± 0.82 | 29.32 ± 24.98 |
| RD (NeuriPS'21) | 10% | 66.44 ± 11.48 | 67.44 ± 15.89 | 36.38 ± 20.99 | 1.72 ± 0.79 | 29.29 ± 22.14 |
| ACMT (MedIA'23) | 10% | 67.13 ± 10.25 | 71.21 ± 13.27 | 38.52 ± 20.43 | 1.65 ± 0.72 | 31.97 ± 23.17 |
| ICT (NN'22) | 10% | 67.14 ± 10.12 | 69.77 ± 14.36 | 36.74 ± 21.96 | 1.70 ± 0.75 | 30.35 ± 23.99 |
| UAMT (MICCAI'19) | 10% | 67.57 ± 11.11 | 68.43 ± 14.28 | 37.83 ± 21.94 | 1.75 ± 0.78 | 28.48 ± 21.51 |
| EM (CVPR'19) | 10% | 67.62 ± 11.16 | 70.44 ± 14.77 | 39.69 ± 26.38 | 1.64 ± 0.72 | 33.16 ± 25.72 |
| UGMCL (AIIM'23) | 10% | 67.81 ± 11.97 | 68.12 ± 15.44 | 38.81 ± 22.52 | 1.69 ± 0.76 | 25.60 ± 22.09 |
| CML (ACMMM'24) | 10% | 68.99 ± 14.16 | 74.75 ± 13.46 | 30.99 ± 17.71 | 1.57 ± 0.78 | **20.26** ± 27.27 |
| MT (NeuriPS'17) | 10% | 67.10 ± 9.24 | 66.69 ± 14.88 | 40.39 ± 20.38 | 1.68 ± 0.71 | 31.21 ± 22.10 |
| **MT + TAR** | 10% | 70.07 ± 10.75 | 69.19 ± 14.93 | 33.83 ± 24.79 | 1.55 ± 0.70 | 37.13 ± 28.37 |
| DyCON (CVPR'25) | 10% | 68.23 ± 14.37 | 68.00 ± 18.10 | 38.68 ± 19.50 | 1.63 ± 0.96 | 21.81 ± 21.98 |
| **DyCON + TAR** | 10% | **74.26** ± 12.25 | **75.52** ± 15.03 | **28.54** ± 20.71 | **1.32** ± 0.81 | 24.16 ± 24.87 |
| Supervised Baseline | 20% | 72.72 ± 11.24 | 71.34 ± 14.43 | 34.72 ± 34.03 | 1.54 ± 0.65 | 32.99 ± 25.86 |
| ADV (BMVC'18) | 20% | 74.00 ± 10.28 | 73.99 ± 12.22 | 31.42 ± 20.48 | 1.45 ± 0.63 | 26.09 ± 22.15 |
| CPS (CVPR'21) | 20% | 74.85 ± 10.88 | 72.65 ± 13.25 | 31.63 ± 27.35 | 1.51 ± 0.73 | 33.22 ± 23.97 |
| RD (NeuriPS'21) | 20% | 74.65 ± 11.45 | 72.16 ± 13.31 | 30.10 ± 26.74 | 1.38 ± 0.72 | 32.23 ± 25.62 |
| ACMT (MedIA'23) | 20% | 73.73 ± 10.66 | 71.99 ± 13.27 | 30.90 ± 23.86 | 1.54 ± 0.61 | 31.88 ± 23.26 |
| ICT (NN'22) | 20% | 73.94 ± 10.72 | 71.43 ± 12.71 | 28.55 ± 22.43 | 1.47 ± 0.61 | 29.22 ± 21.46 |
| UAMT (MICCAI'19) | 20% | 74.21 ± 10.27 | 73.01 ± 12.78 | 30.98 ± 26.56 | 1.46 ± 0.62 | 31.57 ± 22.01 |
| EM (CVPR'19) | 20% | 73.18 ± 9.90 | 73.28 ± 12.40 | 34.99 ± 26.69 | 1.42 ± 0.64 | 27.72 ± 23.04 |
| UGMCL (AIIM'23) | 20% | 74.42 ± 10.97 | 71.21 ± 14.63 | 26.36 ± 20.73 | 1.53 ± 0.65 | 32.19 ± 24.99 |
| CML (ACMMM'24) | 20% | 74.14 ± 16.23 | 77.56 ± 15.68 | 30.03 ± 28.02 | 1.36 ± 0.96 | 32.66 ± 38.35 |
| MT (NeuriPS'17) | 20% | 74.68 ± 10.62 | 72.34 ± 12.91 | 28.08 ± 23.66 | 1.45 ± 0.68 | 31.49 ± 23.63 |
| **MT + TAR** | 20% | 75.21 ± 10.42 | 73.49 ± 13.36 | 27.33 ± 21.68 | 1.39 ± 0.66 | 29.36 ± 22.62 |
| DyCON (CVPR'25) | 20% | 74.20 ± 12.04 | 74.32 ± 16.03 | 27.69 ± 19.91 | 1.32 ± 0.70 | **16.83** ± 13.14 |
| **DyCON + TAR** | 20% | **76.81** ± 13.55 | **77.86** ± 13.44 | **22.50** ± 18.96 | **1.22** ± 0.76 | 22.09 ± 25.02 |
| Supervised Upperbound | 100% | 79.00 ± 11.11 | 76.19 ± 14.22 | 24.07 ± 32.43 | 1.19 ± 0.67 | 24.69 ± 21.87 |

Table 2: Ablation experiments of different components of topological-aware regularization on DyCON semi-supervised segmentation framework using 10% labeled data.

| Method | Dice [%] | clDice [%] | RAVD [%] | ASD[voxel] | 95HD[voxel] | Iteration time[s] |
|---|---|---|---|---|---|---|
| Baseline w/o TAR | 68.23 ± 14.37 | 68.00 ± 18.10 | 38.68 ± 19.50 | 1.63 ± 0.96 | 21.81 ± 21.98 | 0.37 |
| BettiMatching | 61.47 ± 15.44 | 62.55 ± 20.32 | 46.63 ± 24.23 | 1.91 ± 0.90 | 22.46 ± 18.28 | 9.20 |
| clDice | 69.39 ± 16.22 | 69.83 ± 16.95 | 34.95 ± 19.67 | 1.58 ± 1.01 | **21.09** ± 23.38 | 0.51 |
| Skel | 71.75 ± 12.45 | 72.03 ± 16.30 | 32.34 ± 20.13 | 1.40 ± 0.78 | 22.48 ± 21.57 | 1.09 |
| cl-Skel | 73.73 ± 10.31 | 74.45 ± 14.12 | 28.21 ± 20.06 | 1.27 ± 0.64 | 25.74 ± 24.20 | **0.40** |
| clDice+BettiMatching | 65.00 ± 13.65 | 68.39 ± 16.84 | 42.33 ± 22.67 | 1.75 ± 0.86 | 23.74 ± 17.27 | 9.30 |
| clDice+Skel | 72.77 ± 12.30 | 72.62 ± 14.92 | 31.45 ± 21.60 | 1.39 ± 0.74 | 22.14 ± 21.51 | 1.26 |
| clDice+cl-Skel (Ours) | **74.26** ± 12.25 | **75.52** ± 15.03 | **28.54** ± 20.71 | **1.32** ± 0.81 | 24.16 ± 24.87 | 0.79 |

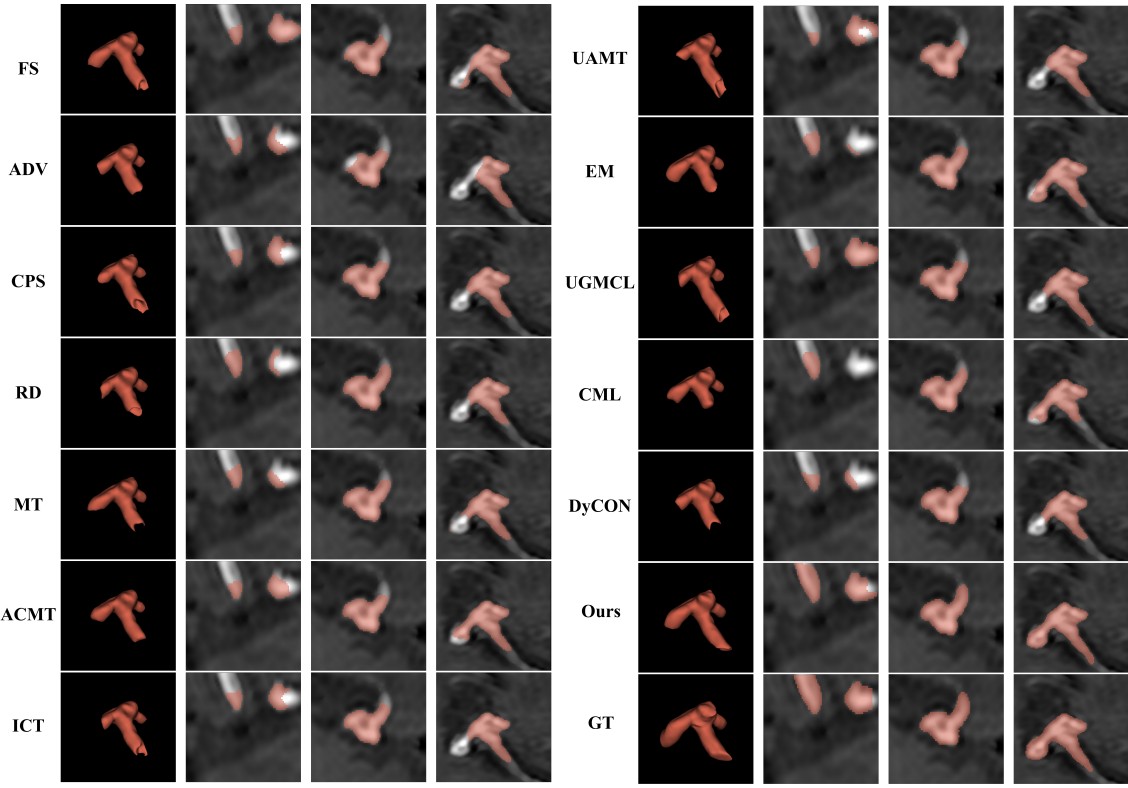

Figure 2: Visual comparison of the intracranial aneurysm vessel results of our proposed method with comparison to fully supervised baseline and other state-of-the-art semi-supervised methods.

### 4.4. Ablation analysis

To validate the contribution of each component within TAR and demonstrate the critical impact of our efficient implementation, we conducted a thorough ablation analysis with comparison to other topological-aware regularization strategies (Stucki et al., 2024) in Table. 2. The results validate that our selection of utilizing both the clDice loss and the Skeleton loss (Skel) achieves the best performance on most metrics including Dice and RAVD. However, the original CPU-based skeletonization (Skel) introduces a severe computational bottleneck, more than doubling the iteration time. The proposed regularization, leveraging the differentiable soft-skeleton algorithm from clDice (cl-Skel), concurrently improves segmentation accuracy and computational efficiency.

## 5. Conclusion and Discussion

In this study, we proposed and validated the core hypothesis that explicitly incorporating topological structure priors is crucial for semi-supervised IA-Vessel segmentation, with experimental results providing strong support. Our Topological-Aware Regularization (TAR), a plug-and-play component, demonstrated significant performance improvements when integrated with different semi-supervised frameworks, proving its efficiency in guiding the model to learn the intrinsic structure of vascular networks. Our findings reveal a key limitation of current semi-supervised learning methods. As shown in the experiments, several classic methods relying on region-based consistency perform even worse than the supervised baseline trained solely on labeled data when handling complex vascular structures. Our work advocates for a paradigm shift from generic, task-agnostic consistency regularization to semantic constraints integrated with specific anatomical priors.

Although our proposed method achieves significant improvements, it still has limitations. Firstly, the effectiveness of TAR remains partially dependent on the quality of pseudo-labels generated by the teacher model. When labeled data is scarce, incorrect topological structures might be treated as supervisory signals, thus affecting the student model's learning. Besides, incorporating more complex topological descriptors into the consistency learning framework could enable more comprehensive structural preservation (Lux et al., 2024; Stucki et al., 2024). Finally, we will explore applying the TAR framework to other medical image segmentation tasks involving tubular or network-like structures to validate its generality and effectiveness (Yao et al., 2024; Tan et al., 2022; Sun et al., 2023; Xue et al., 2020). As a preliminary exploration of integrating topological structure priors into semi-supervised segmentation, our work lays a foundation and provides a new perspective for addressing complex anatomical segmentation tasks with limited labeled data.

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
