# OpenReview forum: "Topological-Aware Regularization for Semi-Supervised Intracranial Aneurysm Vessel Segmentation"
_MIDL.io/2026/Conference — MIDL 2026 Poster_

### Official Review · Reviewer_Q1g5 · 2025-12-29

**Confidence:** 4
**Preliminary Rating:** 4

**Summary:**

This article proposed a topological-aware regularization for semi-supervised intracranial aneurysm vessel segmentation. The mean teacher framework was integrated as semi-supervised backbone. The method was evaluated on a large-scale multi-center MAR dataset and showed SOTA performance. The ablation study approved the bones of the proposed components.

**Strengths:**

The proposed topological-aware regularization loss includes two parts: the L_Skel enforces coverage of the ground-truth centerline, the L_clDice measures topological consistency between prediction and ground truth.

**Weaknesses:**

1.The value of λ1 and λ2 is not clear, which may lead to the irreproducibility.
2.It is not clear which part of the method corresponds to the topological-aware regularization component.
3.The authors evaluated the DyCON network. Considering that DyCON is also a mean-teacher–based architecture and benefits from big loss component, it is unclear how the proposed components and loss functions were implemented in DyCON and the loss component.

**Detailed Comments:**

1.The figure1 needs to be improved, so the reader could understand directly which parts corresponds the Topological-Aware Regularization component.

2.The IAVS dataset is a multi-center collection MRA dataset. When the authors subsampled the data from training set with specific ratios (5%, 10% or 20%), did the authors consider the proportion of data from different center?

3. The conclusion that pixel-/feature-level consistency leads to misleading supervision for tasks with strong structural priors appears to be somewhat speculative, as only one experiment does not clearly establish this causal relationship.

4.The value of λ1 and λ2 is not clear, which may lead to the irreproducibility.
5.It is not clear which part of the method corresponds to the topological-aware regularization component.
6.The authors evaluated the DyCON network. Considering that DyCON is also a mean-teacher–based architecture and benefits from big loss component, it is unclear how the proposed components and loss functions were implemented in DyCON and the loss component.

**Justification Of The Preliminary Rating:**

The paper addresses an important labeling-cost issue in medical imaging and shows reasonable performance gains. However, several concerns limit a stronger recommendation. With clearer explanations, the work could make a solid contribution.

**Questions To Address In The Rebuttal:**

The conclusion that pixel-/feature-level consistency leads to misleading supervision for tasks with strong structural priors appears to be somewhat speculative, as only one experiment does not clearly establish this causal relationship.

---

> ### Author Response · Authors · 2026-01-24
> **Response to Reviewer Q1g5**
>
> We sincerely thank the reviewer for the time and effort dedicated to reviewing our manuscript. We have carefully addressed each of your comments below and revised the manuscript accordingly.
>
> **1. On Hyperparameter Values ($\lambda_1$ and $\lambda_2$)**
>
> **Re:** We clarify that the weights $\lambda_1$ and $\lambda_2$ were both set to 1 in all experiments. In the revised manuscript, we will add a new ablation study on these coefficients.
>
> **2. On Defining the Topological-Aware Regularization (TAR) Component**
>
> **Re:** Thank you for your question. In the revised manuscript, we have clearly defined the specific components of TAR. TAR is not merely a single loss function, but a comprehensive module integrating feature extraction and constraints, consisting of the following three core components:
>
> **Differentiable soft skeletonization algorithm**: Used to extract skeleton features from predicted maps and pseudo-labels in a real-time, differentiable manner during training;
>
> **Skeleton integrity loss**: Designed to penalize skeleton breaks;
>
> **Centerline Dice loss**: Focused on the overall topological overlap.
>
> Together, these three components form the TAR module, which regularizes the student model's learning by aligning topological structures.
>
> **3. On Implementation within DyCON**
>
> **Re:** The TAR we propose is a plug-and-play module. We augment the original DyCON loss function with the TAR loss. Specifically, the DyCON loss function is
> $L_{Total} = L_{Dice} + L_{CE} + \eta \cdot (L_{UnCL} + L_{FeCL})$,
>
> where $L_{sup} = L_{Dice}+L_{CE}, L_{DyCon}=L_{UnCL}+L_{FeCL}$.
>
> When the TAR is added, the total loss becomes: $L_{Total} = L_{Dice} + L_{CE} + \eta \cdot (L_{UnCL} + L_{FeCL}) + L_{Topo}$
>
> **4. On Figure 1 Clarity**
>
> **Re:** We'll revise Figure 1 to enhance its visual clarity. The Topological-Aware Regularization componentis will explicitly highlighted using distinct color-coded blocks and clear labeling, ensuring that its role and data flow within the overall framework are immediately recognizable. These improvements allow the reader to easily distinguish our core topological contribution from the standard segmentation backbone.
>
> **5. On Data Partitioning Strategy**
>
> **Re:** Regarding the data partitioning strategy, we employed a random sampling approach to divide the training, validation, and testing sets, as well as to select the labeled subsets (5%, 10%, and 20%). We did not manually enforce fixed sampling proportions for each specific center.
>
> **6. On Speculative Conclusions regarding Consistency**
>
> **Re:** We appreciate the reviewer’s cautious assessment regarding the generalizability of this conclusion. We clarify that our observation that pixel-level consistency can be misleading is specifically grounded in the context of vascular structures with high topological complexity.
>
> While our current experiments demonstrate that incorporating structural priors significantly improves both segmentation performance and theoretical CFD suitability, we agree that a broader causal relationship requires further investigation. We have revised the manuscript to frame this as a key finding specific to the IA-vessel segmentation task and have explicitly noted in the Future Work section that more extensive cross-domain validation and downstream CFD simulations will be conducted to further substantiate this hypothesis.

---

### Official Review · Reviewer_J21x · 2026-01-07

**Confidence:** 4
**Preliminary Rating:** 3
**Final Rating:** 4

**Summary:**

The paper introduces a semi-supervised segmentation framework for blood vessels with aneurysms, utilizing a regularization strategy grounded in topological correctness even when working with limited labeled data. The proposed regularization method is architecture-agnostic, allowing it to be integrated into various existing segmentation pipelines to improve anatomical continuity

**Strengths:**

I like the aproach to topological regularization, which is adaptable to other frameworks. Furthermore, the focus on semi-supervised learning addresses a major bottleneck while still achieving high-quality segmentation results.

**Weaknesses:**

-The framing of the manuscript makes it difficult to distinguish the primary contributions. Specifically, the interplay between the topological regularization strategy and the student-teacher semi-supervised framework is not clearly articulated, leading to confusion regarding the core novelty.

-Section 2.2 lacks a concrete description of the model architecture. By presenting the methodology in a manner more akin to "background information" rather than a detailed technical specification, it does not seem reproducible

-There is a significant discrepancy between the authors' claims and the data presented in the tables. Specifically, the text asserts that the integration of the proposed module with the DyCON framework (Assefa et al., 2025) establishes a new state-of-the-art. However, the results in the table do not support this conclusion.

**Detailed Comments:**

-The authors state that adding TAR to the DyCON framework "establishes a new state-of-the-art." Could the authors explicitly point to the specific metrics and rows in the table that support this? If the current table is incomplete or contains errors, please provide the corrected data that justifies this claim.

-To enhance readability, please add citations for each of the comparative methods listed in Tables 1 and 2. This will clarify exactly which previous works are being benchmarked against the proposed framework.

**Justification Of Final Rating:**

The core contribution is highly relevant and technically sound for the challenge of vascular segmentation, and the focus on semi-supervised learning with limited annotations is of high practical value. I appreciate the clarifications provided in the rebuttal, which address several of my concerns regarding the student–teacher architecture and experimental setup. However, I remain somewhat skeptical about parts of the reported results

**Justification Of The Preliminary Rating:**

The core contribution is highly relevant and technically sound for the challenge of vascular segmentation. The focus on semi-supervised learning with limited labels is of high practical value to the medical imaging community. However, the current manuscript suffers from  and clarity issues, particularly regarding the student-teacher architecture, the reproducibility of the experiments and inconsistencies between the text and the reported data. If the authors can clarify the distinction between their contributions and provide the missing technical details in a revision, this would be a strong paper.

**Questions To Address In The Rebuttal:**

-To ensure reproducibility, authors should provide a comprehensive description of the network architectures used for both the student and teacher models and not rely entirely on a figure

-The highlighting in Table 1 is misleading. Bold text should consistently denote the top-performing method for each metric, yet the current formatting does not follow a clear convention, which obscures the comparative performance.

---

> ### Author Response · Authors · 2026-01-24
> **Response to Reviewer J21x**
>
> We sincerely thank the reviewer for the time and effort dedicated to reviewing our manuscript. We have carefully addressed each of your comments below and revised the manuscript accordingly.
>
> **1. On the Distinction of Contributions**
>
> **Re:** Thank you for pointing out this critical issue. In the revised manuscript, we have reorganized the methodology section to clearly define the relationship between the foundational framework and our core contribution. Specifically, Mean Teacher (and its variants such as DyCON) serves as the foundational semi-supervised learning backbone, primarily responsible for leveraging unlabeled data distribution information through perturbation consistency. In contrast, the topological-aware regularization proposed in this paper is an independent, plug-and-play core module that imposes additional structural-level consistency constraints beyond pixel-level consistency, by introducing differentiable skeletonization operations and a topological loss function.
>
> **2. On Methodology Description and Reproducibility**
>
> **Re:** We clarify that the primary innovation of this work lies in the semi-supervised learning strategy (i.e., how to effectively utilize unlabeled data) rather than a novel network backbone. Following standard practice in semi-supervised research, the performance gains are achieved through optimized training objectives with exactly the same model (number of parameters or computational complexity) during inference. Our proposed strategy is architecture-agnostic and can be integrated into various different frameworks. In our experiments, we employed a 3D U-Net as our backbone, as specified in the Implementation Details. Our dataset, code, and trained models will be made publicly available upon the formal acceptance of this manuscript to ensure reproducible.
>
> **3. On Discrepancy between Claims and Results**
>
> **Re:** We apologize for any confusion caused by the table and have carefully verified the experimental results. As shown in Table 1, the DyCON framework integrated with TAR (i.e., the row labeled "DyCON + TAR") significantly outperforms the original DyCON and other comparative methods on the vast majority of key metrics. Our method achieves state-of-the-art (SOTA) performance under the current experimental setup in Dice, RAVD, ASD and a new metric clDice.
>
> **4. On Citations in Tables**
>
> **Re:** We fully agree with your suggestion. In the revised manuscript, we have added corresponding reference citations for all comparative methods listed in Tables 1 and 2.
>
> **5. On Network Architecture Details**
>
> **Re:** This network employs a standard 3D U-Net architecture, featuring four downsampling and four upsampling operations, with encoder channel counts of 64, 128, 256, 512, and 1024, respectively. During the decoding phase, skip connections are used for upsampling, and Dropout layers with a dropout rate of 0.3 are added before the bottleneck layer and the final output to prevent overfitting. We have updated these details in Section 4.2, "Implementation Details," of the paper.
>
> **6. On Table Formatting**
>
> **Re:** Thank you for your correction; such informal formatting indeed affects the objectivity of the comparison. We have thoroughly revised the formatting of Tables 1 and 2, with bold text used solely to indicate the optimal metric in each column.

---

### Official Review · Reviewer_3S7Y · 2026-01-09

**Confidence:** 5
**Preliminary Rating:** 4
**Final Rating:** 4

**Summary:**

This paper presents a topology-aware regularization (TAR) method for semi-supervised intracranial aneurysm vessel segmentation from MRA. The proposed approach is lightweight and plug-and-play, explicitly promoting topological correctness by incorporating differentiable skeleton-based losses alongside region-based consistency constraints. Experiments conducted on a multi-center MRA dataset under varying annotation ratios demonstrate consistent performance gains over strong semi-supervised baselines.

**Strengths:**

- Problem formulation grounded in downstream application needs.

- While clDice has been introduced in prior topology-preserving segmentation work, its integration with skeleton integrity constraints in a semi-supervised setting is sensible.

- The ablation studies demonstrate that the proposed method achieves a reasonable trade-off between performance and efficiency.

- The design claims to be working on both Mean Teacher and DyCON frameworks.

- Strong and comprehensive experimental design and evaluation.

**Weaknesses:**

- Dependence on pseudo-label quality remains a concern.
- No downstream evaluation (i.e., mesh-generation) is included.
- Statistical significance analysis is clearly missing to support the reported performance gains.
- Qualitative results are limited, with insufficient topology-focused visual analysis.
- Ambiguity in the definition of supervised and unsupervised losses.

**Detailed Comments:**

- When labeled data is extremely scarce, incorrect topology in pseudo-labels may be reinforced rather than corrected. This limitation is acknowledged but not deeply analyzed.
- Although motivated by CFD usability, the paper does not include a downstream CFD or mesh-success evaluation. Surface metrics are still remain indirect proxies.
- The choice of loss weighting is not deeply discussed.
- Additional visual should emphasis on connectivity errors.

**Justification Of Final Rating:**

While the authors correctly note that downstream CFD evaluation is not strictly required for a segmentation-focused study, the manuscript explicitly motivates the method by its suitability for CFD and mesh generation. In this context, topology-sensitive metrics provide a reasonable proxy, but the link to downstream success remains inferential rather than demonstrated for the present data and error regimes. As a result, the rebuttal improves justification and scope clarity, but the level of empirical evidence supporting downstream readiness remains limited, which constrains overall confidence.

**Justification Of The Preliminary Rating:**

The proposed method represents a solid, well-executed, and practically meaningful contribution to semi-supervised medical image segmentation. The identified weaknesses are relatively minor and should be easily addressed during the rebuttal phase through clearer explanations and clarifications of the existing results, without requiring additional experiments.

**Questions To Address In The Rebuttal:**

Please check the weakness and detailed comments section.

---

> ### Author Response · Authors · 2026-01-24
> **Response to Reviewer 3S7Y**
>
> We sincerely thank the reviewer for the time and effort dedicated to reviewing our manuscript. We have carefully addressed each of your comments below and revised the manuscript accordingly.
>
> **1. Dependence on pseudo-label quality remains a concern..**
>
> **Re:** Thank you for your comment. While semi-supervised learning inevitably utilizes pseudo-labels for training, this method adopts the Mean Teacher architecture, whose exponential moving average strategy effectively suppresses random noise in the predictions of the teacher model, thereby ensuring the relative stability of pseudo-labels. More importantly, the proposed Topology-Aware Regularization (TAR) does not solely rely on pixel-level pseudo-label accuracy. Instead, it focuses on the overall topological connectivity of blood vessels through a differentiable skeleton extraction algorithm. This mechanism enhances the model's robustness to local pixel-level pseudo-label noise, thereby reducing excessive reliance on the absolute quality of pseudo-labels. Experimental results show that even with very few labels (e.g., 5%), this regularization can still effectively improve model performance, demonstrating its adaptability to fluctuations in pseudo-label quality.
>
> **2. No downstream evaluation (i.e., mesh-generation) is included.**
>
> **Re:** We agree that downstream evaluation is valuable and consider it a priority for our future work. However, implementing a full pipeline for mesh generation and CFD involves complex post-processing that falls outside the scope of this study, which focuses on segmentation methodology. According to relevant literature [1], the topological integrity of vessel segmentation (e.g., preventing fragmentation or false fusions) is the primary factor determining the success rate of mesh generation and the accuracy of hemodynamic simulations. Our method directly addresses this "pain point" by significantly improving structural connectivity through explicit topological constraints. Therefore, the substantial gains in topology-sensitive metrics like clDice provide strong evidence that our segmentation results are well-suited for reliable downstream CFD applications.
>
> [1] Pirola S. A Topology-Aware Deep Learning Approach for Automated Multi-Class Segmentation of the Circle of Willis in Modeling Applications. 2025.
>
> **3. Statistical significance analysis is clearly missing to support the reported performance gains.**
>
> **Re:** Thank you for your professional suggestions. In the revised manuscript, we have supplemented the standard deviation data for all quantitative experimental results in the tables to provide a more comprehensive assessment of the performance stability of each method across different data splits. Statistical results show that our method not only achieves improvements in average metrics such as Dice and clDice, but also maintains stable standard deviations across multiple experiments, demonstrating that the performance enhancement of our model is both significant and reliable.
>
> **4. Qualitative results are limited, with insufficient topology-focused visual analysis.**
>
> **Re:** Thank you for pointing this out. We will update the visualization analysis in the paper to focus on the topological structure, making it easier to intuitively observe how the TAR module improves the topological structure of vascular segmentation.
>
> **5. Ambiguity in the definition of supervised and unsupervised losses.**
>
> **Re:** We have further clarified the definition and composition of the loss functions in the Methods section of the manuscript to eliminate potential ambiguities. Within the Mean Teacher semi-supervised framework adopted in this paper, the supervised loss specifically refers to the segmentation error between the student model's predictions and the ground truth labels on labeled data, comprising both Dice Loss and Cross-Entropy Loss. The unsupervised loss, on the other hand, denotes the consistency constraint computed on unlabeled data, which consists of two components: first, a consistency loss (MSE) between the predicted probability maps of the base student and teacher models; and second, the Topology-Aware Regularization loss (TAR) proposed in this paper, which enforces topological consistency between the two models. The revised manuscript provides rigorous mathematical formulations for the computation targets and procedures of these two types of losses.
>
> **6. The choice of loss weighting is not deeply discussed.**
>
> **Re:** We will add sensitivity ablation experiments for the weight coefficient of $L_{Topo}$, as well as for the weight coefficients $\lambda_1$ and $\lambda_2$ within the $L_{Topo}$ component, in the revised paper.

---

### Official Review · Reviewer_35Kv · 2026-01-10

**Confidence:** 5
**Preliminary Rating:** 3

**Summary:**

This paper explores topology-aware regularization for semi-supervised intracranial aneurysm related segmentation. The topic is relevant and the motivation is reasonable, particularly under limited labeled data settings. However, the novelty is unclear since clDice and skeleton-based constraints have been previously introduced, and their combination appears largely incremental. In addition, the experimental setup and evaluation do not fully align with the claimed task and emphasis on topology preservation, and the use of a private dataset without sufficient description raises reproducibility concerns. Overall, the work is technically sound but requires clearer positioning, stronger justification of novelty, and better-aligned evaluation.

**Strengths:**

1.  The paper addresses an important and practical problem by emphasizing topological consistency in aneurysm-related segmentation, which is often inadequately captured by conventional region-based losses.
2. The integration of topology-aware regularization into a semi-supervised learning framework is well motivated, particularly under limited labeled data scenarios.
3. Experimental results demonstrate potential benefits of the proposed approach in preserving structural integrity in IA-related segmentation tasks.

**Weaknesses:**

Major comments:
1. The Related Work section appears to be incomplete and not sufficiently aligned with the focus of the proposed method. Specifically, the authors mainly discuss prior studies on semi-supervised medical image segmentation, while the cited literature and the corresponding discussion are not well matched or systematically organized.

More importantly, although the proposed method emphasizes topology awareness, the manuscript does not adequately review or discuss existing work on topology-aware modeling in intracranial aneurysm (IA) and vascular analysis. Several relevant studies that incorporate topological constraints, vascular connectivity, or geometric consistency in aneurysm or vessel segmentation/analysis are not sufficiently covered or discussed.

2. The manuscript employs the IAVS dataset, which is not a publicly available dataset. According to the description, the only available information about this dataset appears in another manuscript by the same authors that is currently unpublished and only available on arXiv.

4. The second loss term, which encourages the prediction to cover the ground-truth skeleton, is closely related to skeleton- or centerline-supervised constraints that have been explored in prior work on vessel and tubular structure segmentation. Similar ideas have appeared as auxiliary supervision or regularization to improve connectivity and reduce centerline discontinuities.

Given that clDice already incorporates skeleton-based topology modeling, the novelty and necessity of introducing an additional skeleton-guided loss term remain unclear and should be better justified in relation to existing approaches.

3. The experimental setup raises concerns regarding its practical relevance and task formulation. Specifically, both training and testing are conducted on localized image patches that already contain aneurysms, rather than on full MRA volumes. This setting assumes prior knowledge of aneurysm locations, which is not consistent with real-world clinical scenarios, where aneurysms must first be detected within whole-volume images.

Moreover, the segmentation task focuses on the IA–vessel complex, i.e., the aneurysm together with the adjacent parent vessels, rather than performing a dedicated aneurysm segmentation. As a result, the method does not explicitly segment aneurysms alone, which appears to be inconsistent with the paper title and the stated problem definition.

4. Given that the proposed method strongly emphasizes the importance of topological structure preservation in segmentation, it is reasonable to expect that the evaluation should include metrics that explicitly reflect topological correctness, rather than relying solely on region-based overlap measures.

5. The statement “we proposed and validated the core hypothesis that explicitly incorporating topological structure priors is crucial for semi-supervised vessel segmentation” appears to be misleading and overly generalized.

**Detailed Comments:**

1. There is one typographical error in the sentence: we introduce topological-aware regularization (TAR) "for by" incorporating the learning of local vascular topology...
2. The statement “we proposed and validated the core hypothesis that explicitly incorporating topological structure priors is crucial for semi-supervised vessel segmentation” appears to be misleading and overly generalized.

**Justification Of The Preliminary Rating:**

The paper addresses a relevant problem and proposes a technically feasible framework for incorporating topology-aware regularization into semi-supervised segmentation. However, the novelty of the method is not sufficiently clear, as the proposed losses are combination of existing clDice loss and skeleton-based approaches. In addition, several aspects of the experimental design, evaluation protocol, and dataset description would benefit from further clarification and stronger alignment with the stated objectives. With these issues addressed, the work could be strengthened.

**Questions To Address In The Rebuttal:**

1. The authors are encouraged to expand the Related Work section by:
	1).	Improving the correspondence between cited references and the discussion, and
	2).	Providing a more comprehensive review of existing topology-aware approaches in aneurysm and vascular research, clearly positioning the proposed method with respect to prior work.

2. Relying solely on an unpublished arXiv manuscript for the dataset description is insufficient and raises concerns regarding reproducibility and transparency. The authors are strongly encouraged to include a dedicated dataset description section in the manuscript and clearly clarify the dataset’s availability.

3. Clearly justify the use of aneurysm-centered local patches and discuss the limitations of this setting with respect to real clinical deployment.

4. Clarify the exact segmentation target and revise the title and claims accordingly, or alternatively provide experiments that directly address aneurysm-only segmentation.

5. The novelty of this work requires further clarification, as both clDice and skeleton-based integrity constraints have been introduced in prior studies on topology-aware segmentation. It is currently unclear how the proposed formulation differs from or advances beyond existing approaches that already incorporate skeleton- or connectivity-aware losses.

---

> ### Author Response · Authors · 2026-01-24
> **Response to Reviewer 35Kv Q1-Q4**
>
> We sincerely thank the reviewer for the time and effort dedicated to reviewing our manuscript. We have carefully addressed each of your comments below and revised the manuscript accordingly.
>
> **1. On the Organization of Related Work**
>
> **Re:** We agree that a more focused and systematic review of topology-aware modeling in vascular analysis is essential. In the revised manuscript, we have substantially restructured this section to better align with our proposed method. In Section 2.1, "Intracranial Aneurysm and Vessel Segmentation," we systematically review related work on the task of intracranial aneurysm and vessel segmentation. In Section 2.2, "Semi-Supervised Learning in Medical Imaging," we systematically survey related work on semi-supervised medical image segmentation.
>
> **2. On Dataset Availability**
>
> **Re:** We thank the reviewer for the insightful comment regarding the availability of the Intracranial Aneurysm Vessel Segmentation IAVS dataset. We would like to clarify the background and our plan for data sharing as follows:
>
> **Necessity of the New Dataset**: To the best of our knowledge, there is currently no publicly available dataset for intracranial aneurysm vessel segmentation that meets the high-fidelity standards required for CFD analysis. This gap in the field motivated our previous work (currently cited and available on arXiv), where we constructed the first dataset and benchmark specifically tailored for this task.
>
> **Commitment to Open Science**: We fully agree with the reviewer that reproducibility and accessibility are crucial for the progress of the field. We would like to emphasize that the dataset, code, and trained models will be made publicly available upon the formal acceptance of this manuscript. By releasing these resources, we aim to provide a standardized benchmark that allows other researchers to replicate our results and further advance research in intracranial aneurysm analysis and CFD-based clinical applications.
>
> **3. On the Novelty and Necessity of the Skeleton Loss**
>
> **Re:** We clarify that $L_{clDice}$ and $L_{Skel}$ are complementary rather than redundant. While $L_{clDice}$ focuses on the symmetric matching of topological structures, $L_{Skel}$ specifically enforces the prediction to cover the ground-truth centerline, thereby more aggressively preventing vessel fragmentation—a critical requirement for reliable CFD analysis. This synergy is empirically validated in Table 2, where the combined approach (75.52%) significantly outperforms $L_{clDice}$ alone (69.83%) and $L_{Skel}$ alone (72.03%), demonstrating that both terms are necessary to ensure both structural precision and vascular connectivity.
>
> **4. On Experimental Setup and Clinical Relevance**
>
> **Re:**
>
> **Patch-based Segmentation**: We acknowledge that clinical scenarios require whole-volume analysis. The clinical workflow is designed as a two-stage pipeline: global detection followed by localized segmentation. While the first stage (detection) is relatively less annotation-intensive, the second stage (segmentation) requires immense manual effort for pixel-level labeling. This work focuses on the segmentation task because pixel-level annotation requires significantly more manual effort than detection. The annotation-efficient patch-based segmentation model could be integrated with detection pre-processors to achieve a practical, end-to-end solution for full MRA volumes.
>
> **Clinical Relevance for CFD**: We appreciate the reviewer’s discussion on clinical relevance. Our task formulation is strictly tailored for downstream CFD simulation requirements. For intracranial aneurysm rupture risk assessment, segmenting the aneurysm alone is insufficient. To construct accurate hemodynamic boundary conditions, it is essential to fully segment the aneurysm and its connected parent arteries. Our task definition, Intracranial Aneurysm Vessel Segmentation reflects this clinical necessity.
>
> In the revised paper, we have added the above two points of explanation in Section 3.1, Task Definition.

---

> ### Author Response · Authors · 2026-01-24
> **Response to Reviewer 35Kv Q5-Q7**
>
> **5. On Evaluation Metrics**
>
> **Re:** We agree that region-based metrics like Dice are insufficient to reflect topological correctness. In the revised manuscript, we have integrated **clDice** into all experimental results (Table 1 and Table 2). As a metric specifically designed to evaluate skeleton-level topological consistency, clDice provides a more objective and rigorous validation of our method’s effectiveness in preserving vascular connectivity compared to traditional overlap-based measures.
>
> **6. On the Hypothesis Statement**
>
> **Re:** We appreciate the reviewer’s rigorous feedback. In the revised manuscript, we have refined this claim to “we proposed and validated the core hypothesis that explicitly incorporating topological structure priors is crucial for semi-supervised IA-Vessel segmentation.”.
>
> **7. On Typographical Errors**
>
> **Re:** We sincerely apologize for this typographical error. We have removed the redundant "for" from the text. Thank you for your careful observation.

---

> > ### Comment · Reviewer_35Kv · 2026-02-01
> >
> > We sincerely thank the authors for their comprehensive and thoughtful responses to our concerns. The authors have demonstrated a high level of academic integrity and commitment to improving the manuscript. We appreciate the substantial revisions made, particularly the reorganization of related work, the addition of clDice as a topology-sensitive evaluation metric, and the commitment to publicly release the dataset, code, and trained models upon acceptance.

---

### Author Rebuttal · Authors · 2026-01-25

**Rebuttal:**

Dear Area Chair and Reviewers,

We sincerely thank you for your constructive comments, which have significantly helped us improve the quality and rigor of our work. Please note that as we cannot directly update the submission PDF, the revised manuscript has been uploaded to the **"Supporting Material"** section. We kindly ask you to refer to this updated version during the discussion.

In response to your feedback, we have made the following key improvements to the manuscript:

1. **Topology-Aware Metric**: We have integrated clDice into all experimental results (Table 1 & 2) to strictly evaluate the preservation of vascular connectivity.

2. **Statistical Analysis**: We added standard deviations for all quantitative experiments to demonstrate the stability and statistical significance of our method.

3. **Clinical Justification**: We expanded Section 3.1 to clarify that segmenting the full IA-Vessel complex is a prerequisite for downstream CFD analysis.

4. **Presentation & Clarity**: We restructured the Related Work section for better positioning, added citations for all comparative methods, and will improve Figure 1 to visually distinguish the TAR module.

We have provided detailed responses to each reviewer’s specific questions in the individual threads below. We reiterate our commitment to Open Science and confirm that the IAVS dataset, source code, and model weights will be made publicly available upon acceptance to ensure reproducibility. We hope these revisions address your concerns and look forward to further discussion.

**Supporting Material:**

/attachment/12c20ec72e6a4f13068ef7fc467d821110c6bd6e.pdf

---

### Comment · Area_Chair_CQvN · 2026-01-26
**Post-rebuttal discussion and final ratings**

Dear reviewers,

Thank you for providing your comments on the paper. The authors have replied, and possibly modified their paper as a result.

You now have until the 1st of February to discuss with authors via the forum, and provide your final rating. This can change or stay the same depending on the rebuttal and discussion.

Your review, discussion, and final rating will be taken into account for the meta-review.

Once again thank you very much for your help.

---

### Comment · Area_Chair_CQvN · 2026-02-02
**Please provide final rating today**

Dear reviewers, the rebuttal and discussion periods are over, could you please submit your final ratings today if not done already?

To do so please use "edit" on your official review.

Thank you again for your precious help in evaluating the papers.

---

### Meta-Review · Area_Chair_CQvN · 2026-02-05

**Recommendation:** Accept (Poster)
**Confidence:** 4

**Metareview:**

All reviewers judged the submission positively and some improved their rating post rebuttal.
The authors have improved the paper post rebuttal and provided clear answers.

---

### Decision · Program_Chairs · 2026-02-13

Accept (Poster)